# Relationships Between Effort, Rate of Perceived Exertion, and Readiness During a Warm-Up for High-Speed Sprinting

**DOI:** 10.3390/jfmk10020155

**Published:** 2025-05-01

**Authors:** Roland van den Tillaar, Nick Ball, Per Øyvind Torvik

**Affiliations:** 1Department of Sports Sciences and Physical Education, Nord University, 7600 Levanger, Norway; per.o.torvik@nord.no; 2Faculty of Health, Research Institute for Sport and Exercise Science, University of Canberra, Canberra 2601, Australia; nick.ball@canberra.edu.au

**Keywords:** masters, speed, warm-up, running velocity

## Abstract

**Objectives**: The aim of this study was to investigate how a sprint warm-up with increasing prescribed effort relates to actual effort and how this influences RPE and readiness for a maximal 50 m sprint performance. **Methods**: A total of 19 subjects (17 men and 2 women age: 43.8 ± 12.6 yrs., height: 1.78 ± 0.08 m, body mass: 78.7 ± 9.5, 100 m PB: 13.07 ± 1.0) undertook a short specific warm-up of 8 × 50 m runs with 60 s rest in between (10 min in total) where a dynamic exercise was performed. The first 50 m run was performed at a self-estimated effort of around 60% of estimated maximal sprint speed. Each subsequent 50 m repetition required a 5% increase in sprint speed until it reached 95% of maximal self-estimated intensity, followed by a maximal 50 m sprint performance. Every 50 m time was measured together with the rating of perceived exertion (RPE) and readiness to perform a maximal 50 m sprint. **Results**: The main findings were that actual percentage of effort was generally higher than prescribed efforts, especially in the initial test, while alignment improved in the retest, except at higher intensities (80–90%). Furthermore, both RPE and readiness had a significant positive correlation with the percentage of effort, though RPE was consistently lower, and readiness was slightly reduced at lower efforts in the retest. In addition, test–retest reliability indicated consistent sprint performance and perceptual measures across sessions. **Conclusions**: It was concluded that this short, structured warm-up is suitable for maximal sprint performance as shown by the readiness and RPE.

## 1. Introduction

Many different warming up strategies have been used that consist of both general and sport specific elements, which has been reported to regulate motivation [1], improve concentration and perceptual awareness [2], prevent injuries [3,4], and enhance athletic readiness and performance [5,6]. It is suggested that a general warm-up elevates body temperature, promotes cardiovascular activation, and provides a broad physical base (e.g., joint mobility, movement readiness) [7,8], while specific warm-ups mimic the movements of the target activity to fine tune neuromuscular coordination and readiness [9,10]. In fast explosive movements, such as sprinting, it is important to be specifically “prepared” sufficiently to avoid possible injuries [11,12] and to enhance explosive muscular [13,14] and sprint performance [10] compared to just general warm-ups.

Warm-up duration has been the subject of much discussion with Bishop [7] indicating that often the duration of the warm-up is too long. van den Tillaar and colleagues [10,15,16] found that a short specific warm-up consisting of 8 × 40–60 m sprints with increasing effort (65–95%), interspersed with dynamic stretches were as effective as longer warm-ups on sprint, repeated sprint, and intermediate running performance (e.g., 3-min maximal run). Furthermore, athletes within these studies reported that the short warm-up was less exhausting than the long warm-up [10,15,16]. As warm-ups are designed to prepare the athlete for competition both physically and psychologically, understanding the athletes perceived readiness for maximal effort during the warm-up would allow further understanding on warm-up length and intensity efficacy. Ibbott et al. [17] considered ‘readiness to lift’ in their studies on self-paced inter-set rest interval during a resistance training task indicating that as efforts progressed, readiness decreased. Within warm-ups, coaches can prescribe intensities/effort with the intent to build up the athlete to peak athletic preparation. However, in all of the aforementioned studies, the prescribed effort was not investigated in relation to the perceived or actual effort performed for each sprint run. This could be much different for each individual and provide insight into the progressive intensity of the warm-up. It is not currently known whether subjects meet the expected effort intensity during a progressive warm-up. In addition, it is unclear how perceived exertion (RPE), and feelings of readiness evolve over the course of different phases of the warm-up. Additionally, post-warm-up readiness to sprint has not been measured. Based on the potential for individual responses to the same warm-up intervention, understanding the progressive differences in warm-up intensity and its relationship to the readiness to perform maximal explosive activity would be of benefit to practitioners.

Therefore, the objective of the study was to examine how a structured warm-up with increasing effort relates to actual effort and how this affects RPE, pre-sprint readiness, and performance in 50 m sprints. Unlike previous studies, which assess RPE only after warm-up, our study measures it at each sprint. Furthermore, the study examines whether these relationships remain consistent across repeated tests in a random sub-set of participants. Within this study, we have mainly used master athletes as they have years of training experience and many face age-related declines in physical capacity. These age-related physiological and biomechanical changes necessitate a precise and effective warm-up to ensure peak performance and injury prevention and to minimize fatigue. Furthermore, the number of competing master athletes has increased in recent years, alongside improvements in performance [18]; however, not much is known about the effects of such a short warm-up in master athletes. We hypothesize that prescribed and actual efforts will correlate strongly, that RPE and readiness will increase with effort, and that training experience will improve alignment between prescribed and actual effort while reducing RPE at lower intensities across repeated tests [18]. These findings have the potential to inform the development of more efficient, effective, and individualized warm-up protocols that enhance both physical and psychological preparedness for sprint performance.

## 2. Materials and Methods

### 2.1. Participants

A total of 19 participants (17 men and 2 women, age: 43.8 ± 12.6 yrs., height: 1.78 ± 0.08 m, body mass: 78.7 ± 9.5 kg, 100 m PB: 13.07 ± 1.0) participated in the present study. All participants were active athletes, and some were competing at the international master athletics championships. Mostly master athletes were used as they have much experience with warm-ups and need normally a longer warm-up before performing a maximal sprint than younger athletes [19]. Written informed consent was obtained from all participants. Permission to conduct the study was obtained from the University of Canberra Ethics Committee. (HREC -1772) and complied with the current ethical standards in sports and exercise science research.

### 2.2. Procedure

Each participant was asked to avoid heavy resistance or sprint training for 24 h prior to each testing session and to maintain the same routine in the 24 h period prior to each test (if they tested twice). The tests were conducted on an indoor tartan running track with a temperature of 23–25 °C, and participants were allowed to wear their spikes during each sprint and on both days. Each participant performed at least one test session, while eleven participants completed two repeat sessions to establish intra-subject reliability. The test session consisted of a short specific warm-up of 8 × 50 m runs with 60 s rest in between (10 min in total). During the 60 s rest, subjects completed a dynamic mobility exercise. The first 50 m was performed at a self-estimated intensity/effort of around 60% of estimated maximal sprinting velocity/effort. Following this, each subsequent 50 m effort was increased by 5% until it reached 95% of maximal self-estimated intensity/effort on the 8th repetition. In each rest period, one of the seven dynamic exercises for the shoulders (arm swing), hip (internal/external, hip rotations, extension/flexion and abduction/adduction), knee (rotations), and ankle joints (joint rotations) were administered in accordance with prior protocols by van den Tillaar et al. [10] and for further details. These dynamic exercises were performed 10 times each to increase the range of motion in each joint and to avoid the heart rate decreasing much in this period. The protocol was a modified version of earlier studies in which 60 m distances in sprint [10,15,16] or 100 m in cross country skiing [20] were used as distances to warm up with. After the eight runs, a maximal 100% effort 50 m sprint was performed with 3 min of active rest (easy walking) before. Most of the participants did not have any previous experience with exactly this warm-up protocol, but almost all participants were familiar with warm-up sprints with increasing intensity as part of their warm-up for competition.

### 2.3. Measurements

Each 50 m sprint effort was measured by 2 pairs of double beam electronic photocell timing gates (Swift Performance Equipment, Wacol, Australia), linked to an electronic timer. The participants started in a 2-point split stance behind a line 0.3 m behind the first beams, which were placed at a height of 92.5 cm (top beam) and 68 cm (bottom beam). The last pairs of beams were placed at the 50 m mark. Time on each sprint was measured. To calculate the actual percentage of maximal effort for each run, the sprint times for each warm-up repetition were divided by the 50 m time during the maximal 50 m sprint.

Readiness was asked no more than 10 s prior to the start of the next run by the question ‘If you have to run a maximal 50 m sprint now, how ready are you for it on a 0–10 scale in which 0 is not ready at all, while ten is give me the maximal 50 m sprint now’. In the context of this study, when asking this question, the individual was informed to consider overall readiness, considering both their physical and psychological state at that time point, to undertake a maximal effort. Although this parameter was not validated before in this setting, we think that this parameter could give a good indication about how much warm-up runs per participant are necessary before performing at maximal effort. Following each run, the received perception exertion (RPE) was asked on a Borg scale of 0–10 in which 0 indicated no exertion and 10 indicated maximal perceived exertion [21].

Eleven male participants were selected at random to perform repeat testing four days after the first test to assess the intra-subject reliability parameters. These participants undertook the same protocol and were tested at the same time of day.

### 2.4. Statistical Analysis

Normality of data distribution was assessed and confirmed using the Shapiro–Wilk test. Test–retest reliability for all variables was evaluated using Cronbach’s intra-class correlation coefficient (ICC). To assess the development of the sprint times and perceptual variables (RPE and readiness) during each test a one-way analysis of variance (ANOVA) with repeated measurements on each parameter was used. The percentage of maximal effort was tested against the prescribed percentage of maximal effort by a paired sampled *t*-test at each described effort percentage. To compare the test–retest results, a 2 (test, retest) × 9 (sprint effort 60–100%) repeated ANOVA was applied on each parameter. Post hoc comparisons with Holm–Bonferroni corrections were conducted to locate differences. All results are presented as mean ± SD. Where sphericity assumptions were violated, Greenhouse–Geisser adjustments of the *p*-values were reported. The alpha level was set at 0.05. Pearson’s correlation was used for evaluating the association between readiness, RPE, and percentage of effort for each athlete at each test day. A correlation ≤ 0.29 was considered trivial, 0.3–0.49 weak, 0.5–0.69 moderate, between 0.70 and 0.89 was considered high, and over 0.9 was considered very high [22]. Effect size was evaluated with η^2^ (partial ETA squared), where 0.01 < η_p_^2^ < 0.06 constitutes a small effect, 0.06 < η_p_^2^ < 0.14 constitutes a medium effect, and η_p_^2^ > 0.14 constitutes a large effect [23]. Statistical analysis was performed in JASP 19.1 (University of Amsterdam, Amsterdam, The Netherlands).

## 3. Results

As a group, the sprint time from the prior effort was always slower than the subsequent effort. Large variations occurred in the first run (40–90%) compared to efforts (~97%) in the final run of the warm-up (F ≥ 56.4, *p* < 0.001, η_p_^2^ ≥ 0.85, Figure 1).

When comparing the prescribed percentage of effort with the actual percentage, a significant difference was found. In the first test, all actual percentages were significantly higher than the prescribed percentage (*p* ≤ 0.029, η_p_^2^ ≥ 0.15) except at 60% of maximal effort. At the retest only, at 80–90% of maximal efforts, the athletes ran significantly faster (*p* ≤ 0.029, η_p_^2^ ≥ 0.15) than prescribed (Figure 2).

Furthermore, a significant effect of run was found for readiness and RPE in both the test and retest (F ≥ 54.1, *p* < 0.001, η_p_^2^ ≥ 0.76, Figure 3). The athletes increased readiness significantly each run until 90% of maximal effort at the test. At the retest it increased significantly in every run. For RPE also in each run, this significantly increased from the previous run (Figure 3).

When comparing readiness and RPE between the test and retest, a significant interaction effect was found for readiness (F = 2.47, *p* = 0.025, η_p_^2^ = 0.20) and RPE (F = 8.72, *p* < 0.001, η_p_^2^ = 0.47). Post hoc comparison revealed that readiness was significantly higher at early runs (65 and 70%) during the test compared to the retest. Additionally, a significant interaction effect (F = 5.7, *p* = 0.039, η_p_^2^ = 0.36) between the test and retest was found. Post hoc comparison revealed significantly higher RPE during the first three runs during the test compared to retest (Figure 3). No other significant test or interaction effects were found for sprint time, percentages, and perceptual parameters (F ≤ 0.62, *p* ≥ 0.57, η_p_^2^ < 0.01).

Significant high correlations were found between percentage of sprinting effort, readiness, and RPE at both test and retest (Table 1). Also, the ICCs between the test and retest for these parameters were of acceptable consistency [24].

## 4. Discussion

The aims were to investigate how a short progressive sprint-based warm-up with increasing prescribed effort relates with actual effort and how this subsequently influences RPE and readiness for a maximal 50 m sprint performance. The main findings were that actual percentage of effort was generally higher than prescribed efforts, especially in the initial test, while alignment improved in the retest, except at higher intensities (80–90%). Furthermore, both RPE and readiness had a significant positive correlation with the percentage of effort, though RPE was consistently lower, and readiness was slightly reduced at lower efforts in the retest. In addition, test–retest reliability indicated consistent sprint performance and perceptual measures across sessions.

The overestimation of actual versus prescribed effort, especially in the first test, can be explained by several factors. Athletes initially overestimated their pace (while some underestimated the effort in the first sprints, Figure 1) due to incomplete internal calibration of effort relative to prescription. Over time, this calibration improved with experience as shown during the retest. Even with improved familiarity during the retest, athletes may still face challenges in accurately gauging their effort at high intensities (80–90%) due to the nonlinear nature of physiological and perceptual scaling at near-maximal efforts (Figure 2). At these intensities, small discrepancies in pacing or physiological readiness can lead to significant performance variations. It is shown that athletes are less precise in matching perceived effort to actual output as they approach maximal effort due to the greater involvement of anaerobic systems and rapid onset of fatigue [12]. The literature on pacing strategies supports the notion that repeated exposure to structured efforts refines internal effort regulation, improving alignment with targets [5,8]. Familiarization sessions might mitigate this effect, aligning initial prescribed and actual efforts more closely. Furthermore, as the point of the warm-up was to increase effort each time with approximately 5%, it is easy to increase a bit more than 5% from time to time. Thereby, over the different sprints, this overestimation can result in overestimation at the later runs (80–90%) as they have to increase effort each time, before effort cannot increase much more as it becomes a near maximal or maximal sprint (95 and 100%).

During the different sprints with increasing efforts, the Rate of Perceived Exertion (RPE) and readiness increased similarly as shown by the very large correlations between effort with RPE and readiness. This is explainable by the fact that session RPE reflects the body’s internal monitoring of physiological strain, such as heart rate, oxygen consumption [25], and muscular exertion [26]. As effort increases, these physiological demands also increase, leading to a proportional rise in perceived exertion. Research consistently show that RPE scales linearly with physical effort intensity, especially during structured and progressive activities [25]; however, some research claims that RPE is less accurate at high intensity [27]. This relationship is well established in the Borg RPE scale framework [21]. On the other hand, at high intensity, objective physiological measures are considered more valid and reliable then RPE [25].

It is more interesting to see that readiness also increased with increasing effort (Figure 3) and that already after the 85% effort run, four (test) and two (retest) of the athletes marked readiness as a 9 or higher, while after the 90% effort sprints, this was increased to eleven (test) and seven (retest). During the first test, this readiness did not increase significantly anymore after 90% (Figure 3), thereby indicating that most athletes already were ready to perform maximal effort sprints after just 6–7 50 m sprints (85–90% effort).

The RPE and readiness development changed from the first to second test, where in the first test RPE and readiness were higher during the 60–70% effort sprints, causing an interaction effect and test effect (readiness) (Figure 3). The main reason for the lower RPE and readiness at the first runs was that the athletes paced themselves better with less overestimation at low efforts (Figure 2) during the retest due to the previous exposure to this protocol. A repeated exposure to a warm-up protocol leads to improved neuromuscular efficiency and energy economy [28]. Whether or not the familiarity with the tasks reduces the physiological cost of submaximal efforts, resulting in a lower RPE at lower intensities during the retest, is relatively unexplored in the literature.

Furthermore, psychological familiarity with the protocol can reduce cognitive and emotional stress [5], diminishing perceived difficulty [1], thereby lowering perceived effort. Due to the better pacing, the RPE and readiness are following a more linear correlation with effort during the retest compared with the first test, which indicates a learning/experience effect.

Therefore, this also indicates one of the limitations of the study, which shows that familiarization to a new type of warm-up protocol can be of importance before testing it, which might have reduced initial discrepancies between prescribed and actual efforts. Furthermore, only eleven male athletes were included in the retest, which is a small sample size, especially when no women were included. This limits generalizability across sexes. Moreover, in the present study mainly master athletes were used due to their training experience and possibly longer necessary expected duration for warm-ups. So, findings may not generalize to younger or elite athletes. However, as shown, most athletes were ready to perform maximal sprint runs after just 6–7 sprints with increasing effort indicating that even for older athletes this type of warm-up is efficient enough for maximal sprint performance. Whilst ‘readiness’ has been assessed prior to maximal efforts in prior studies, the statement used in this paper had not been validated prior; as such we acknowledge that the statement only takes into account the athletes readiness on that day, and aspects that contribute to the readiness external to the warm-up protocol are not accounted for (i.e., sleep quality, diet, muscle soreness, etc.). However, as shown by the ICC of 0.71 on readiness between test and retest, it seems that the parameter has a good limit of agreement, but more studies on this parameter should be performed to confirm this. Lastly, the absence of kinematic analyses prevents insight into biomechanical changes that occur during this warm-up, which contribute to performance differences. Future studies could address these gaps by including larger, more diverse samples, incorporating kinematic assessments, and testing protocols across varied athletic populations.

## 5. Conclusions

It was concluded that actual percentage of effort was generally higher than prescribed efforts, especially in the initial test, while alignment improved in the retest, except at higher intensities (80–90%). Furthermore, both RPE and readiness had a significant positive correlation with the percentage of effort and that test–retest reliability indicated consistent sprint performance and perceptual measures across sessions. The study highlights the importance of a short, structured warm-up protocol with incremental efforts for optimizing readiness and performance. Findings underscore the need for individualized strategies that accommodate learning effects, effort calibration, and psychological preparedness. Based upon the present finding, we suggest that this short, structured warm-up is suitable for maximal sprint performance as shown by the readiness and RPE and that individual adjustments (less sprints with increasing effort) should be made so each athlete can perform maximally during their sprint performance.

## Figures and Tables

**Figure 1 jfmk-10-00155-f001:**
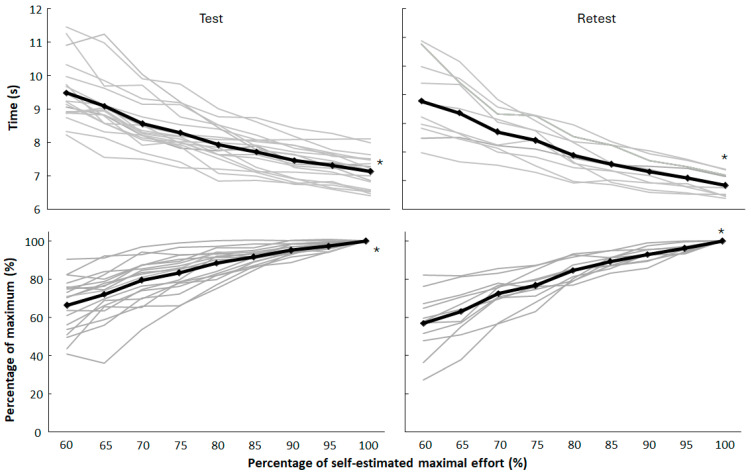
Mean (black) and individual (grey) sprint times and percentage of maximal effort per run over the whole protocol for test and retest. * Indicates a significant increase between each run.

**Figure 2 jfmk-10-00155-f002:**
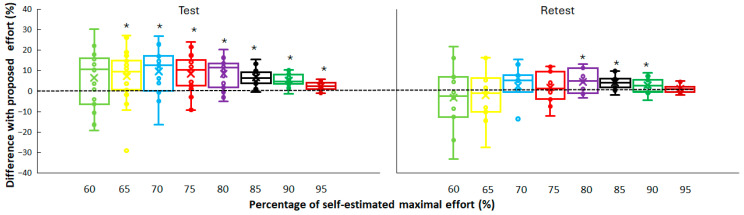
Mean (x), median, 95% confidence intervals, and individual difference in percentage with the prescribed percentage of effort in test and retest. * Indicates a significant overestimation at this percentage of self-estimated maximal effort.

**Figure 3 jfmk-10-00155-f003:**
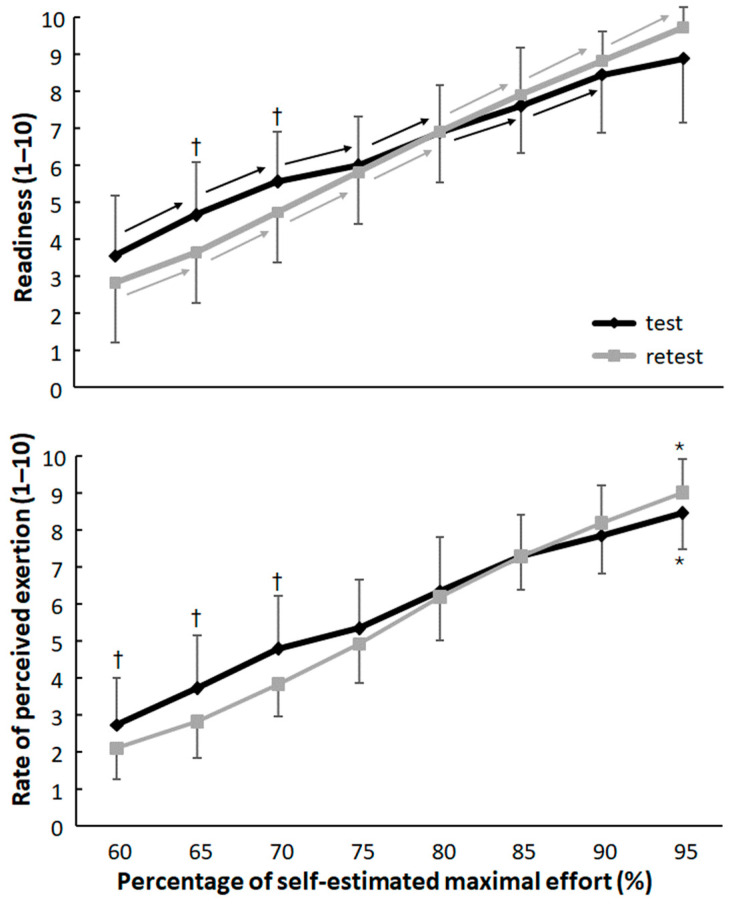
Average (±SD) readiness and rate of perceived exertion (RPE) at each run during the test and retest. * Indicates a significant increase between each run for this test. → Indicates a significant increase with the next run for this test. † Indicates a significant difference between the test and retest for this parameter at this run.

**Table 1 jfmk-10-00155-t001:** Correlations between percentage of effort, readiness, and RPE at test and retest together with intra-class correlations (ICC) between test and retest over all athletes.

	Test		Retest		
	Readiness	RPE	Readiness	RPE	ICC
Percentage of effort	0.72	0.81	0.80	0.78	0.73
Readiness	-	0.81		0.90	0.71
RPE					0.75

## Data Availability

The data presented in this study are available upon request from the corresponding author. The data are not publicly available due to the national laws of the Norwegian government regarding privacy.

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
