# Peer review of "Relationships Between Effort, Rate of Perceived Exertion, and Readiness During a Warm-Up for High-Speed Sprinting"

_jfmk, 2025, doi:10.3390/jfmk10020155_

Round 1
Reviewer 1 Report
Comments and Suggestions for Authors
Title. I think you can replace the title, like: “Relationships Between Effort Intensity, RPE, and Readiness During Warm-Up Before Maximal 50m Sprints”.
Introduction
Line 33. The phrase “with claims that it can regulate motivation” is vague. Consider revising for clarity and academic tone: “which has been reported to regulate motivation” or “warm-up has been shown to regulate motivation”.
Line 34. The term “awareness” is broad. You might specify whether you mean “situational awareness”, “perceptual awareness”, or “alertness”.
Line 36. “Broad physical base” is vague. Consider clarifying what physical qualities this includes (e.g., mobility, joint preparation, or general movement readiness).
Line 41. The infinitive “to enhance” is grammatically disconnected. Consider rephrasing: “which are intended to enhance” or “designed to improve”.
Line 43. Typo/grammar error: “has been source of much discussion” should be corrected to: “has been the subject of much discussion.”
Line 45. Phrase “8 times 40–60m sprints” should follow standard formatting: 8 × 40–60m sprints” for clarity.
Lines 48-49. Multiple spelling errors: intenisties = intensities, intrent = intent, preperation = preparation.
Lines 53-54. Sentence structure is unclear. Suggested rephrasing: “Moreover, RPE has been assessed only after the warm-up, not during its progression. Additionally, post-warm-up readiness to sprint has not been measured.”
Lines 59-61. The aim is well-articulated but could be split into two sentences for clarity: “The present study aimed to examine how a structured warm-up with increasing effort relates to actual effort performed. It also explored how this influences RPE, readiness, and subsequent sprint performance.”
Lines 65-68. This final sentence is strong. You might consider specifying how findings may benefit coaches or applied settings: “may help coaches develop more effective and individualized warm-up strategies.”
Methods
The methodology is generally sound and aligns with the study’s aim. The use of master athletes is justified based on training experience and age-related considerations. However, there are several areas where clarity and detail should be improved. The description of the dynamic exercises is too vague and should be supplemented with a specific list or referenced protocol for reproducibility. The warm-up protocol is clearly structured, but certain elements—such as the rationale for using a 5% incremental increase in effort or how “prescribed effort” was explained to participants—need further elaboration. The subjective readiness scale, while innovative, is not validated and its interpretation may vary. Additionally, only 11 participants completed the retest, all male, which limits the generalizability of reliability results and should be acknowledged as a methodological limitation. Finally, the statistical analysis is mostly appropriate, though it would benefit from clarification on how assumptions (e.g., normality, sphericity) were tested and whether effect sizes were interpreted according to standard guidelines.
Line 75. “and potentially fatigues”. Grammatically incorrect and unclear. Likely a typo. Suggest: “and to minimize fatigue”.
Lines 75-76. “the number of master athletes competing… increases with also increasing performances”. Awkward phrasing. Suggested revision: “the number of competing master athletes has increased in recent years, alongside improvements in performance.”
Line 80. Include measurement unit for body mass (e.g., kg).
Lines 92-93. Well described. You may consider specifying environmental conditions if available (e.g., temperature, humidity), as they could influence sprint performance.
Discussion
The discussion provides a generally coherent interpretation of the results and places the findings within the context of relevant literature. The distinction between RPE and readiness, and their correlation with effort, is well presented. However, the discussion lacks depth in several key areas. The term “readiness” is not conceptually well defined—whether it refers to physiological, neuromuscular, or psychological preparedness is unclear. Furthermore, the lack of objective physiological or biomechanical measures limits the strength of the conclusions. While the authors acknowledge the small and male-only sample in the retest, its impact on the robustness of reliability findings is underexplored. The discussion would also benefit from a clearer connection to practical applications for coaches or athletes. Lastly, individual differences in pacing and perception, as well as the learning effect noted in the retest, could be more deeply analyzed to highlight the variability in response to the protocol.
Comments on the Quality of English Language
The manuscript requires substantial editing for grammar, syntax, and academic tone. There are numerous spelling and typographical errors, inconsistent verb tenses, and informal expressions that reduce clarity and readability. A thorough review by a native English speaker or a professional editing service is strongly recommended to ensure the manuscript meets publication standards.
Author Response
We want to thank the reviewer for the comments on the manuscript. We have now changed the manuscript in accordance with the comments of the reviewer and think that it is now suitable for publication.
Reviewer 1
Title. I think you can replace the title, like: “Relationships Between Effort Intensity, RPE, and Readiness During Warm-Up Before Maximal 50m Sprints”.
We have changed the title in: Relationships between effort intensity, rate of perceived exertion and readiness during warming up before maximal 50m sprints according to the comments of reivewers1 and 2.
Introduction
Line 33. The phrase “with claims that it can regulate motivation” is vague. Consider revising for clarity and academic tone: “which has been reported to regulate motivation” or “warm-up has been shown to regulate motivation”.
We have changed it in: … has been reported to regulate motivation
Line 34. The term “awareness” is broad. You might specify whether you mean “situational awareness”, “perceptual awareness”, or “alertness”.
Changed it in perceptual awareness
Line 36. “Broad physical base” is vague. Consider clarifying what physical qualities this includes (e.g., mobility, joint preparation, or general movement readiness).
We have included: (e.g. joint mobility, movement readiness) to the text.
Line 41. The infinitive “to enhance” is grammatically disconnected. Consider rephrasing: “which are intended to enhance” or “designed to improve”.
Changed it according to the comment of the reviewer.
Line 43. Typo/grammar error: “has been source of much discussion” should be corrected to: “has been the subject of much discussion.”
Changed it according to the comment of the reviewer.
Line 45. Phrase “8 times 40–60m sprints” should follow standard formatting: 8 × 40–60m sprints” for clarity.
Changed it according to the comment of the reviewer.
Lines 48-49. Multiple spelling errors: intenisties = intensities, intrent = intent, preperation = preparation.
Thank you for spotting this. We did not see this and have changed it accordingly.
Lines 53-54. Sentence structure is unclear. Suggested rephrasing: “Moreover, RPE has been assessed only after the warm-up, not during its progression. Additionally, post-warm-up readiness to sprint has not been measured.”
Changed it according to the comment of the reviewer.
Lines 59-61. The aim is well-articulated but could be split into two sentences for clarity: “The present study aimed to examine how a structured warm-up with increasing effort relates to actual effort performed. It also explored how this influences RPE, readiness, and subsequent sprint performance.”
Changed it according to the comment of the reviewer.
Lines 65-68. This final sentence is strong. You might consider specifying how findings may benefit coaches or applied settings: “may help coaches develop more effective and individualized warm-up strategies.”
We have included this to the text now.
Methods
The methodology is generally sound and aligns with the study’s aim. The use of master athletes is justified based on training experience and age-related considerations. However, there are several areas where clarity and detail should be improved. The description of the dynamic exercises is too vague and should be supplemented with a specific list or referenced protocol for reproducibility. The warm-up protocol is clearly structured, but certain elements—such as the rationale for using a 5% incremental increase in effort or how “prescribed effort” was explained to participants—need further elaboration. The subjective readiness scale, while innovative, is not validated and its interpretation may vary. Additionally, only 11 participants completed the retest, all male, which limits the generalizability of reliability results and should be acknowledged as a methodological limitation.
- Dynamic Exercises – The exercises are taken from a prior publication that used this protocol.
van den Tillaar, R., Lerberg, E., & von Heimburg, E. (2019). Comparison of three types of warm-up upon sprint ability in experienced soccer players. Journal of Sport and Health Science, 8(6), 574-578.
These exercises have been included on line 106-109
- Rationale for using 5% -
The prior papers by Van der Tillaar et al used the arbitrary 5% increased protocol used in this paper. This paper, as stated, aimed to ascertain whether the 5% increase aligned with the athletes perceived effort. Thus the paper aims to try and support the use of this verbal protocol.
- Subjective Readiness Scale
The readiness question was included to ascertain a specific subjective indication of their ability to perform maximally. Such a question has been used prior by authors in resistance training studies whereby ‘Readiness to Lift’ was used as an indicator of pacing strategies in resistance training.
Ibbott, P., Ball, N., Welvaert, M., & Thompson, K. G. (2019). The effect of self-paced and prescribed interset rest strategies on performance in strength training. International Journal of Sports Physiology and Performance, 14(7), 980-986.
As the prescribed 5% increment here requires a form or tele-anticipatory quality seen in pacing, we felt this question would help us understand how much of the warm-up was needed from the subjects perspective before they felt ‘ready’ to sprint maximally. We do however acknowledge that this isn’t validated as such and therefore internal factors such as sleep quality, muscle soreness etc are not quantifiable from the response to the readiness question. We have included a line in the limitations to indicate this.
- Method Limitation
this is included – line 254
Finally, the statistical analysis is mostly appropriate, though it would benefit from clarification on how assumptions (e.g., normality, sphericity) were tested and whether effect sizes were interpreted according to standard guidelines.
We have included this to the text now under statistics. Parts of it were already mentioned under this part.
Line 75. “and potentially fatigues”. Grammatically incorrect and unclear. Likely a typo. Suggest: “and to minimize fatigue”.
We have changed it according to the comments of the reviewer.
Lines 75-76. “the number of master athletes competing… increases with also increasing performances”. Awkward phrasing. Suggested revision: “the number of competing master athletes has increased in recent years, alongside improvements in performance.”
We have changed it according to the comments of the reviewer.
Line 80. Include measurement unit for body mass (e.g., kg).
Included now.
Lines 92-93. Well described. You may consider specifying environmental conditions if available (e.g., temperature, humidity), as they could influence sprint performance.
We have included the temperature that we had to the text. Humidity we did not measure.
Discussion
The discussion provides a generally coherent interpretation of the results and places the findings within the context of relevant literature. The distinction between RPE and readiness, and their correlation with effort, is well presented. However, the discussion lacks depth in several key areas. The term “readiness” is not conceptually well defined—whether it refers to physiological, neuromuscular, or psychological preparedness is unclear. Furthermore, the lack of objective physiological or biomechanical measures limits the strength of the conclusions. While the authors acknowledge the small and male-only sample in the retest, its impact on the robustness of reliability findings is underexplored. The discussion would also benefit from a clearer connection to practical applications for coaches or athletes. Lastly, individual differences in pacing and perception, as well as the learning effect noted in the retest, could be more deeply analyzed to highlight the variability in response to the protocol.
Thank you for the question. The readiness question was a simple way to help us understand the current perceived state of the individual just prior to the effort. It was explained at the beginning that this question was to consider all aspects associated with readiness including physical and psychological. This question had a projection aspect where the athlete had to consider how they may be able to perform given their current state. This question was asked 10s prior to the effort. The RPE was taken directly after the effort to get a gauge of the exertion level.
In this paper, a similar protocol was used in relation to repeat effort back squats (https://journals.sagepub.com/doi/epub/10.1177/0031512519835976) whereby ready to lift opposed to ready to sprint maximally was used, with RPE recorded post the effort.
Comments on the Quality of English Language
The manuscript requires substantial editing for grammar, syntax, and academic tone. There are numerous spelling and typographical errors, inconsistent verb tenses, and informal expressions that reduce clarity and readability. A thorough review by a native English speaker or a professional editing service is strongly recommended to ensure the manuscript meets publication standards.
The manuscript is proofread again by a native English speaker and we think that it is now up to the standards of the journal.
Reviewer 2 Report
Comments and Suggestions for Authors
Dear authors,
I have reviewed your manuscript entitled:
Relationships Between Effort Intensities, RPE, Readiness During Warming Up For Maximal 50m Sprints
The study aims to assess how a structured sprint warm-up with increasing prescribed effort intensities aligns with actual effort and how this impacts RPE (Rating of Perceived Exertion) and readiness for a maximal 50m sprint.
After finalizing the revision of the manuscript, I would like to make a few comments:
- Abstract: The Keywords must be improved. Consider adding more keywords.
- Materials and Methods: Measurements (lines 118,119, 120) - Readiness was asked no more than 10s prior to the start of the next run by the question: 118 ‘If you have to run a maximal 50m sprint now, how ready are you for it on a 0-10 scale in which 0 119 is not ready at all, while ten is give me the maximal 50m sprint now’. Bibliographic reference is missing! How reliable is this questionnaire?
- Results: At least a table with the most important results could help the reader's reading and understanding!
- Discussion: The discussion could be improved, particularly about readiness. There is still a lack of bibliographical references regarding this studied variable!
The study presents valuable findings and is methodologically sound but requires minor revisions for clarity and depth in analysis. I recommend Minor Revisions before considering the manuscript for publication.
Once again, I would like to congratulate you on your work. I hope you will consider all my comments.
Best regards
Author Response
We want to thank the reviewer for the comments on the manuscript. We have now changed the manuscript in accordance with the comments of the reviewer and think that it is now suitable for publication.
I have reviewed your manuscript entitled:
Relationships Between Effort Intensities, RPE, Readiness During Warming Up For Maximal 50m Sprints
The study aims to assess how a structured sprint warm-up with increasing prescribed effort intensities aligns with actual effort and how this impacts RPE (Rating of Perceived Exertion) and readiness for a maximal 50m sprint.
After finalizing the revision of the manuscript, I would like to make a few comments:
- Abstract: The Keywords must be improved. Consider adding more keywords.
We have included more keywords.
- Materials and Methods: Measurements (lines 118,119, 120) - Readiness was asked no more than 10s prior to the start of the next run by the question: 118 ‘If you have to run a maximal 50m sprint now, how ready are you for it on a 0-10 scale in which 0 119 is not ready at all, while ten is give me the maximal 50m sprint now’. Bibliographic reference is missing! How reliable is this questionnaire?
In our opinion it is a similar question to RPE. However, it is a subjective feeling about how ready you feel if you would have to run maximal now. Since the athletes are experienced with their feeling for competition preparation, we expected that they would have a high constancy in reporting this. We have now included ICC analysis to show if this was the case.
- Results: At least a table with the most important results could help the reader's reading and understanding!
We cannot report both figures and table on the same number as it is called double reporting. We think that the figures show much better the development over the different intensities than a table would give. That is why we have chosen for figures and not tables. We have included a table with the correlations (which we forgot to include to the text). We hope the reviewer appreciates are point of view. For exact numbers evt. interested viewers could contact the authors.
- Discussion: The discussion could be improved, particularly about readiness. There is still a lack of bibliographical references regarding this studied variable!
We have added a reference about this.
Reviewer 3 Report
Comments and Suggestions for Authors
Introduction: You should restructure the introduction by relating background, problem, and objective in a clearer and more straightforward manner.
Problem: You should make the problem more rational and related to the research objective.
Objective: To investigate how a structured warm-up protocol with increasing effort relates to actual effort and how this affects RPE, readiness and performance on the 50 metre sprint.
Redefine the problem by linking it better with the objective
Method: The method used is relevant to the objective. However, some changes must be made in certain sections.
LAWS 75-77 “Furthermore, in later years the number of master athletes competing in athletics increases with also increasing performances over time [17] and not much is known about the effects of such short warm-up in master athletes.” à This definition should be included in the introduction and not in the methods.
Partecipants: Despite reaching the sampling limit, you might add a power analysis so as to verify the significance of the sampling.
Procedure: The test took place in an indoor track and the athletes were allowed to wear spiked shoes. “Each participant performed at least one test session, while eleven participants completed two repeat sessions to establish intra-subject reliability.” à This section is not clear, as only 11 athletes in the group repeated the test for the re-evaluation of the effort made to understand reliability and repeatability. It is not defined according to which criteria only 11 athletes were chosen.
Results: You should rephrase the following passage to make it clearer.
LAWS 142-143 varying from 40-90% in the first run to around 97% in the last run of the warm-up (F ≥ 56.4, p < 0.001, η2 ≥ 0.85, Figure 1). à You should rephrase the following passage to make it clearer.
Discussion: The discussions provide a simple and effective explanation of the results, while remaining relevant to the objectives set.
Conclusion: Conclusions should be more detailed and should verify whether the results, positive compared with the hypothesis, have practical applications. It needs to be further studied and demonstrated how this aspect of self-assessment is an additional tool for knowledge improvement.

Author Response
We want to thank the reviewer for the comments on the manuscript. We have now changed the manuscript in accordance with the comments of the reviewer and think that it is now suitable for publication.
Introduction: You should restructure the introduction by relating background, problem, and objective in a clearer and more straightforward manner.
The introduction has been amended for clarity
Problem: You should make the problem more rational and related to the research objective.
Objective: To investigate how a structured warm-up protocol with increasing effort relates to actual effort and how this affects RPE, readiness and performance on the 50 metre sprint.
This has been included
Redefine the problem by linking it better with the objective
Method: The method used is relevant to the objective. However, some changes must be made in certain sections.
LAWS 75-77 “Furthermore, in later years the number of master athletes competing in athletics increases with also increasing performances over time [17] and not much is known about the effects of such short warm-up in master athletes.” à This definition should be included in the introduction and not in the methods.
This has been moved to the introduction.
Participants: Despite reaching the sampling limit, you might add a power analysis so as to verify the significance of the sampling.
Procedure: The test took place in an indoor track and the athletes were allowed to wear spiked shoes. “Each participant performed at least one test session, while eleven participants completed two repeat sessions to establish intra-subject reliability.” à This section is not clear, as only 11 athletes in the group repeated the test for the re-evaluation of the effort made to understand reliability and repeatability. It is not defined according to which criteria only 11 athletes were chosen.
We have included that the participants were randomly selected
Results: You should rephrase the following passage to make it clearer.
LAWS 142-143 varying from 40-90% in the first run to around 97% in the last run of the warm-up (F ≥ 56.4, p < 0.001, η2 ≥ 0.85, Figure 1). à You should rephrase the following passage to make it clearer.
The line has been amended.
Discussion: The discussions provide a simple and effective explanation of the results, while remaining relevant to the objectives set.
Thank you we have made some minor amendments to further enhance clarity
Conclusion: Conclusions should be more detailed and should verify whether the results, positive compared with the hypothesis, have practical applications. It needs to be further studied and demonstrated how this aspect of self-assessment is an additional tool for knowledge improvement.
Thank you, the conclusion includes a line indicating the practical aspects of the finding in relation to warm-up length for maximal sprinting.
Reviewer 4 Report
Comments and Suggestions for Authors
title
i suggest write the entire name in the tittle about RPE: rate of perceived effort
introduction:
line 45-47: add for which activy was effective the warm up
Methods
the participants avoid caffeine the test day?
the dynamic exercises in the rest are exercise of mobility ? please describe better in the text.
what was the active rest prior to the maximum sprint ? please describe in the text
The reliability should be analyzed by intraclass correlation coeficient (Call to increase statistical collaboration in sports science, sport and exercise medicine and sports physiotherapy reference). please remake these statistic. you can keep the repeated measures ANOVA analysis with two within-subject factors (test retest factor and sprint factor).
add the lower categorizations (trivial, small): A correlation between 0.5 and 0.69 was considered moderate, between 0.70 and 0.89 was considered high, and over 0.9 was considered very high
change these: line 133 "The criterion level for significance was set at p<0.05" for " the alfa level was set at 0.05"
change the effect size for partial eta squared. The advantage is wich is comparable with other studies (reference: Calculating and reporting effect sizes to facilitate cumulative science: a practical primer for t-tests and ANOVAs)
results
Add all effect size in results (partial eta squared)
correlations results are missing
figure 1: describe the the graph of left and right in description. Is test and retest data ?
figure 2: add * description
Author Response
We want to thank the reviewer for the comments on the manuscript. We have now changed the manuscript in accordance with the comments of the reviewer and think that it is now suitable for publication.
title
i suggest write the entire name in the tittle about RPE: rate of perceived effort
We have changed the title now and written rate of perceived exertion.
introduction:
line 45-47: add for which activity was effective the warm up
This is mentioned in line 48: on sprints, repeated sprints and intermediate running performance (e.g. 3-min maximal run).
Methods
the participants avoid caffeine the test day?
The participants were not avoided to drink caffeine if they did regularly. They were allowed to have their normal drinking routine as we think it would otherwise have a negative influence upon their normal sprinting performance. They were asked to uphold the same routine as normal.
the dynamic exercises in the rest are exercise of mobility ? please describe better in the text.
We have included more explanation in the text. For detailed description it is referred to earlier study of van den Tillaar et al. who also had pictures included.
what was the active rest prior to the maximum sprint ? please describe in the text
Easy walking. This is included to the text now.
The reliability should be analyzed by intraclass correlation coeficient (Call to increase statistical collaboration in sports science, sport and exercise medicine and sports physiotherapy reference). please remake these statistic. you can keep the repeated measures ANOVA analysis with two within-subject factors (test retest factor and sprint factor).
We have now included ICCs for the different parameters based upon test and retest results.
add the lower categorizations (trivial, small): A correlation between 0.5 and 0.69 was considered moderate, between 0.70 and 0.89 was considered high, and over 0.9 was considered very high
We have included these to the text now.
change these: line 133 "The criterion level for significance was set at p<0.05" for " the alfa level was set at 0.05"
Changed now according to the comment of the reviewer.
change the effect size for partial eta squared. The advantage is wich is comparable with other studies (reference: Calculating and reporting effect sizes to facilitate cumulative science: a practical primer for t-tests and ANOVAs)
All effect sizes are given in partial eta squared. Most were already given. We have included the missing ones.
results
Add all effect size in results (partial eta squared)
This is done.
correlations results are missing
Sorry, we had them, but forget to report them. They are now included to the result part.
figure 1: describe the the graph of left and right in description. Is test and retest data ?
Included now in text and figure.
figure 2: add * description
Included now to the figure
Round 2
Reviewer 1 Report
Comments and Suggestions for Authors
Accept in present form.
Author Response
Thank you
Reviewer 3 Report
Comments and Suggestions for Authors
FOREWORD
The manuscript presents an interesting and well-structured study designed to investigate the relationship between prescribed, perceived, and actual effort during a progressive sprint warm-up protocol, with a focus on subjective variables such as RPE and readiness. The topic is relevant to both scientific research and practical application in sports, especially considering the focus on master’s athletes. The methodology adopted is generally sound and the statistical analyses are appropriate with respect to the stated objectives. However, some aspects-such as the presentation of the problem, methodological clarity, depth of discussion, and emphasis on practical applications-could be further improved to strengthen the overall effectiveness and impact of the work. The following comments are made with the intent to support the authors in optimizing the manuscript.
INTRODUCTION: The introduction appears reorganized in a more straightforward manner than the previous version, but some passages can be further simplified and made more direct so as to improve overall clarity. It would be helpful to state the central objective of the study from the outset, preventing it from being confused by an excess of general information.
PROBLEM: The problem underlying the study is present, but it is not expressed clearly enough. Currently, it is embedded within a larger discussion of warm-ups, and this reduces its impact. To make it more obvious, we recommend that it be included as a stand-alone statement, clearly and directly worded. A possible rephrasing could be, “It is not currently known whether subjects meet the expected effort intensity during a progressive warm-up. In addition, it is unclear how perceived exertion (RPE) and feelings of readiness evolve over the course of different phases of the warm-up.”
- Moreover, the Gap with the literature is not explicit, but implicit. It would be helpful to clearly state, “Unlike previous studies, which assess RPE only after warm-up, our study measures it at each sprint.”
- OBJECTIVE: The objective now appears to be clearer and more understandable, as it is clearly spelled out “The objective of the study was to examine how a structured warm-up with increasing effort relates to actual effort and how this affects RPE, readiness, and performance in 50m sprints.” However, it is suggested that the logical transition from general background to goal formulation should be strengthened. Also, it would be helpful to better explain the rationale for choosing the master athlete population in the context of the objective.” The objective appears to be consistent with the problem; in fact, after introducing that it is unclear whether prescribed effort and actual effort coincide, and that RPE and readiness were never measured during the warm-up, the objective to measure these very things appears logical and consistent.
METHODS: The methods section is clearly explained in that the procedures are clear, detailed and relevant to the objectives, but there are some aspects that deserve clarification.
The positive points are that:
the Warm-up protocol is well described:
- 8 sprints of 50 meters with the 5% increment at each reps;
- Indication of initial intensity (60%) and final intensity (95%);
- 60 seconds of recovery between tests, during which mobility exercises are performed, explained for each joint).
Test repetition (retest):
- Clearly indicate that 11 participants repeated the protocol after 4 days and these were randomly selected.
Well-defined measures:
- Actual effort calculated with percentage of time vs. maximum sprint.
- RPE measured after each sprint, readiness before each sprint, with 0-10 scale.
- Use of photocells for time measurement.
Points for clarification and improvement appear to be:
- The protocol is new. It is unclear whether the subjects had had time to familiarize themselves with it or not. So, it is suggested to indicate whether participants had been previously familiarized with the warm-up protocol.
- Subjective measurement tools not validated. The readiness scale is purpose-built and not scientifically validated (although this is admitted to the study).
These clarifications would improve the transparency and replicability of the study.
STATISTICAL ANALYSIS: An INTERCLASS CORRELATION COEFFICIENT was added to the previous statistical analysis, which was already well set up, to assess the reliability of the re-test. However, there continues to be no application of the Pearson correlation presented in the method.
RESULTS: This section appears to be clear. Nevertheless, there is a lack of inclusion of Pearson's correlation, as presented in the method, to make it comprehensive.
DISCUSSION: The discussions overall are well written and correctly interpret the results of the study. The key points emerge clearly, and the language is accessible. However, it is suggested that a more in-depth comparison with the existing literature and expansion of the discussion of methodological limitations, such as the use and validity of a scale for readiness not described in the literature, should be made. These improvements would make the discussion more comprehensive and effective.
CONCLUSIONS: The following section has not been updated as requested by the previous comment “Conclusions should be detailed and should test in summary whether the results, which are positive with respect to the hypothesis and the new self-assessment protocol, have practical applications. It needs to be further studied and demonstrate how this aspect of self-assessment is an additional tool for knowledge improvement.”

Author Response
We have now revised the manuscript according to the comments of the reviewer and think that it is now suitable for publication.
INTRODUCTION: The introduction appears reorganized in a more straightforward manner than the previous version, but some passages can be further simplified and made more direct so as to improve overall clarity. It would be helpful to state the central objective of the study from the outset, preventing it from being confused by an excess of general information.
PROBLEM: The problem underlying the study is present, but it is not expressed clearly enough. Currently, it is embedded within a larger discussion of warm-ups, and this reduces its impact. To make it more obvious, we recommend that it be included as a stand-alone statement, clearly and directly worded. A possible rephrasing could be, “It is not currently known whether subjects meet the expected effort intensity during a progressive warm-up. In addition, it is unclear how perceived exertion (RPE) and feelings of readiness evolve over the course of different phases of the warm-up.”
We have changed this according to the comment of the reviewer.
- Moreover, the Gap with the literature is not explicit, but implicit. It would be helpful to clearly state, “Unlike previous studies, which assess RPE only after warm-up, our study measures it at each sprint.”
We have included this to the text now according to the comment of the reviewer.
- OBJECTIVE: The objective now appears to be clearer and more understandable, as it is clearly spelled out “The objective of the study was to examine how a structured warm-up with increasing effort relates to actual effort and how this affects RPE, readiness, and performance in 50m sprints.” However, it is suggested that the logical transition from general background to goal formulation should be strengthened. Also, it would be helpful to better explain the rationale for choosing the master athlete population in the context of the objective.” The objective appears to be consistent with the problem; in fact, after introducing that it is unclear whether prescribed effort and actual effort coincide, and that RPE and readiness were never measured during the warm-up, the objective to measure these very things appears logical and consistent.
We have changed it according to the comment of the reviewer.
METHODS: The methods section is clearly explained in that the procedures are clear, detailed and relevant to the objectives, but there are some aspects that deserve clarification.
The positive points are that:
the Warm-up protocol is well described:
- 8 sprints of 50 meters with the 5% increment at each reps;
- Indication of initial intensity (60%) and final intensity (95%);
- 60 seconds of recovery between tests, during which mobility exercises are performed, explained for each joint).
Test repetition (retest):
- Clearly indicate that 11 participants repeated the protocol after 4 days and these were randomly selected.
Well-defined measures:
- Actual effort calculated with percentage of time vs. maximum sprint.
- RPE measured after each sprint, readiness before each sprint, with 0-10 scale.
- Use of photocells for time measurement.
Thank you
Points for clarification and improvement appear to be:
- The protocol is new. It is unclear whether the subjects had had time to familiarize themselves with it or not. So, it is suggested to indicate whether participants had been previously familiarized with the warm-up protocol.
Only a few subjects (2) had experience with exactly this protocol. However, most of the athletes have experience in performing several warm-up runs with increasing intensity, but not following exactly this protocol. We have now added: “Most of the participants did not have any previous experience with exactly this warm-up protocol, but almost all participants were familiar with warm-up sprints with increasing intensity as part of their warm-up for competition.” To the text.
- Subjective measurement tools not validated. The readiness scale is purpose-built and not scientifically validated (although this is admitted to the study).
We have added this to the text: “Although this parameter was not validated before in this setting, we think that this parameter could give a good indication about how much warm-up runs per participant are necessary before performing at maximal effort.”
These clarifications would improve the transparency and replicability of the study.
STATISTICAL ANALYSIS: An INTERCLASS CORRELATION COEFFICIENT was added to the previous statistical analysis, which was already well set up, to assess the reliability of the re-test. However, there continues to be no application of the Pearson correlation presented in the method.
In lines 155 and 156 it is written that Pearson’s correlation was used for evaluating the association between readiness, RPE and percentage of effort for each athlete at each test day.
RESULTS: This section appears to be clear. Nevertheless, there is a lack of inclusion of Pearson's correlation, as presented in the method, to make it comprehensive.
In line 195 and table 1 the results of the correlations were shown between percentage of effort with readiness and RPE.
DISCUSSION: The discussions overall are well written and correctly interpret the results of the study. The key points emerge clearly, and the language is accessible. However, it is suggested that a more in-depth comparison with the existing literature and expansion of the discussion of methodological limitations, such as the use and validity of a scale for readiness not described in the literature, should be made. These improvements would make the discussion more comprehensive and effective.
We have include more about readiness in the text in the discussion: However, as shown by the ICC of 0.71 on readiness between test and retest, it seems that the parameter has a good limits of agreement, but more studies on this parameter should be performed confirm this.
CONCLUSIONS: The following section has not been updated as requested by the previous comment “Conclusions should be detailed and should test in summary whether the results, which are positive with respect to the hypothesis and the new self-assessment protocol, have practical applications. It needs to be further studied and demonstrate how this aspect of self-assessment is an additional tool for knowledge improvement.”
We have adapted the conclusions according to the comment of the reviewer.
Reviewer 4 Report
Comments and Suggestions for Authors
Thanks you for considering my suggestions, congratulations for the work.
Author Response
Thank you